

# Origin, burial and preservation of late Pleistocene-age glacier ice in Arctic permafrost (Bylot Island, NU, Canada)

Stephanie Coulombe[1 2 3], Daniel Fortier[2 3 5], Denis Lacelle[4], Mikhail Kanevskiy[5], Yuri Shur[5 6]

[1]Polar Knowledge Canada, Cambridge Bay, X0B 0C0, Canada
[2]Department of Geography, Université de Montréal, Montréal, H2V 2B8, Canada
[3]Centre for Northern Studies, Université Laval, Quebec City, G1V 0A6, Canada
[4]Department of Geography, Environment and Geomatics, University of Ottawa, Ottawa, K1N 6N5, Canada
[5]Institute of Northern Engineering, University of Alaska Fairbanks, Fairbanks, 99775-5910, USA
[6]Department of Civil and Environmental Engineering, University of Alaska Fairbanks, Fairbanks, 99775-5960, USA

Correspondence *to*: Stephanie Coulombe (stephanie.coulombe@polar.gc.ca)

Over the past decades, observations of buried glacier ice exposed in coastal bluffs and headwalls of retrogressive thaw slumps of the Arctic indicate that considerable amounts of Pleistocene glacier ice survived the deglaciation and are still preserved in permafrost. In exposures, relict glacier ice and intrasedimental ice often coexist and look alike but their genesis is strikingly different. Identifying the origin of ground ice is required to model its spatial distribution and abundance, which
is necessary to model the response of circumpolar permafrost regions to climate change. This paper aims to present a detailed description and report physical and geochemical properties of glacier ice buried in the permafrost of Bylot Island (Nunavut) as well as identify geomorphic processes that led to the burial and preservation of the ice. The massive ice exposure and core samples were described according to the cryostratigraphic approach, combining the analysis of permafrost cryofacies and cryostructures, ice crystallography, stable O-H isotopes and cation contents. The buried glacier ice consisted
of clear to whitish englacial ice having large crystals (cm) and small gas inclusions (mm) at crystal intersections, similar to observations of englacial ice facies commonly found on contemporary glaciers and ice sheets. However, the isotopic composition of the buried ice differed markedly from contemporary glacier ice and indicated the late Pleistocene age of the ice. This ice predates the aggradation of the permafrost and can be used as an archive to infer paleo-environmental conditions at the study site. As most of the arctic landscapes are still strongly determined by its glacial legacy, the melting of
these large ice bodies could lead to extensive slope failures and settlement of the ground surface, with significant impact on permafrost geosystem landscape dynamics, terrestrial and aquatic ecosystems, and infrastructure.

## 1 Introduction

In the Arctic, extensive areas of ridged and hummocky moraines are underlain by buried glacier ice (Alexanderson et al., 2002; Dyke and Savelle, 2000; Kokelj et al., 2017; Smith, 2015; Solomatin, 1986). Massive-ice bodies within these
landscapes are important indicators of past glacial, hydrologic and hydrogeologic conditions and are used to reconstruct regional paleo-environments and paleo-climates (Fritz et al., 2011; Murton et al., 2005). Areas with buried glacier ice are





also becoming increasingly vulnerable to climate warming (Kokelj et al., 2015, 2017; Segal et al., 2016). Glacier ice is the most common type of buried ice in permafrost and its occurrence was reported in Russia (Astakhov, 1986; Astakhov et al., 1996; Belova, 2015; Forman et al., 1999; Ingólfsson and Lokrantz, 2003; Kaplanskaya and Tarnogradskiy, 1986; Solomatin, 1986), the Canadian Arctic (Dallimore and Wolfe, 1988; French and Harry, 1990; Hyatt et al., 2003; Moorman and Michel,

2000; St-Onge and McMartin, 1999), Alaska (Jorgenson and Shur, 2008; Kanevskiy et al., 2013), and Antarctica (Astakhov, 1986; Belova, 2015; Dallimore and Wolfe, 1988; Forman et al., 1999; Hyatt et al., 2003; Ingólfsson and Lokrantz, 2003; Moorman and Michel, 2000; St-Onge and McMartin, 1999; Sugden et al., 1995; Swanger, 2017). Buried glacier ice has been commonly observed in the proglacial zone of contemporary glaciers and can be preserved in formerly glaciated areas (the paraglacial to periglacial zones) within large moraine belts, hummocky till and glaciofluvial deposits (Everest and Bradwell,

2003; Senderak et al., 2017; Tonkin et al., 2016). In the Canadian Arctic, buried glacier ice has been mainly described in the western and central regions, especially in the Mackenzie Delta region (French and Harry, 1990), Tuktoyaktuk Coastlands (Murton et al., 2005), Herschel Island (Fritz et al., 2011), central Yukon (Lacelle et al., 2007), Banks Island (Lakeman and England, 2012) and Victoria Island (Dyke and Savelle, 2000). While the permafrost of the eastern Canadian Arctic is expected to contain remnants of Pleistocene ice sheets or glaciers, very few have been reported so far.

Distinguishing between buried glacier ice and other types of massive ground ice in the permafrost is usually based on cryostratigraphy combined with detailed studies of physical, geochemical and isotopic properties of the ice that may also include analyses of occluded gases (Cardyn et al., 2007; Fritz et al., 2011; Ingólfsson and Lokrantz, 2003). When classifying glacier ice facies, a distinction is made between englacial and basal ice facies (Fortier et al., 2012; Hubbard et al., 2009;

Lawson, 1979; Lorrain et al., 1981). The englacial (firn-derived) ice facies is formed by the gradual snow compaction and recrystallization, a process called firnification, and has a low debris content (Benn and Evans, 2010). Basal ice has distinctive physical and chemical characteristics and has a much higher debris content than the overlying englacial ice as a result of subglacial processes operating at the glacier bed: regelation, glaciohydraulic supercooling accretion, net basal adfreezing, incorporation of ice and sediments by overriding ice during glacier advance, glacio-tectonics, ice lens

aggradation and downward propagation of cold temperature in the sediment at the glacier bed (congelation) (Alley et al., 1998; Cook et al., 2006; Evans, 1989; Fortier et al., 2012; Hubbard and Sharp, 1995; Knight, 1997; Lawson, 1979; Sharp et al., 1994). Both types of glacier ice may experience burial but basal ice is probably the most common form of buried glacier ice according to reports from various permafrost regions (Belova et al., 2008; Fritz et al., 2011; Murton et al., 2005; St-Onge and McMartin, 1999). The process of burial of glacier ice has been described by Shur (1988), Solomatin (1986), Harris and

Murton (2005) and citations therein. Burial of glacier ice may occur as a result of (1) accumulation of fluvial, lacustrine, aeolian, or slope sediments on top of the ice; (2) formation of insulating blanket of supraglacial melt-out till (Larson et al., 2016). With this scenario, sediment-rich basal ice has a greater potential to persist in a buried state than englacial ice with little debris; or (3) glaciotectonic processes (Harris and Murton, J. B., 2005). Buried glacier ice remains stable for a long





period of time only if the soil temperature is below freezing, and the active layer thickness does not exceed a depth to the massive ice body (Shur, 1988).

5 In this study, we describe the occurrence of massive ice preserved in the permafrost of the Qarlikturvik Valley, southwestern Bylot Island (NU, Arctic Canada). We investigated the physical and geochemical properties of a recently exposed body of massive ice and compared them with those of other ice types in the region (snow, glacier ice, wedge ice, segregated ice) to infer its origin. A cryostratigraphic approach was used to delineate cryostratigraphic units on the basis of their cryostructures, physical properties, and thaw unconformities (French and Shur, 2010; Gilbert et al., 2016), combined with crystallography and geochemical analyses of the different ground-ice types. The origin of the massive ice is discussed.

## 10  2 Regional Setting

With ice-capped summits dominating the central highlands and glaciated valleys that extend near the coast, the mountainous central section of the island forms a striking contrast with the relatively flat coastal lowlands. The Byam Martin Mountains range (~1400 m a.s.l.) consists primarily of Archean-Aphebian crystalline igneous rocks and Proterozoic metasedimentary and metamorphic rocks (Jackson and Davidson, 1975). Klassen (1993) suggested that alpine glaciers, larger but similar in 15 size to those present today, occupied Bylot Island during the late Wisconsinan. At the last glacial maximum (LGM), ice streams of the Laurentide Ice Sheet (LIS) flowed in adjacent marine channels (Lancaster Sound, Navy Board Inlet, and Eclipse Sound) and reached Bylot Island (De Angelis and Kleman, 2007; Dyke and Hooper, 2001). The study area is situated in the Qarlikturvik Valley (73˚09' N, 79˚57' W, 25 m a.s.l.) on southwest Bylot Island, Arctic Canada (Fig. 1). This valley was eroded through a Cretaceous-Tertiary sequence of poorly consolidated sandstone and shale (Jackson and Sangster, 20 1987).

The Qarlikturvik Valley is a typical U-shaped glacial valley with surface sediments reflecting the complex history of the valley: presence of unconsolidated glacial, colluvial, alluvial, marine, aeolian and organic deposits dating back to the Late Pleistocene and Holocene (Allard, 1996; Fortier et al., 2006; Fortier and Allard, 2004). The valley comprises low-lying ice-25 wedge polygon terraces bordering a proglacial braided river running in a glaciofluvial outwash plain and forming a delta in Navy Board Inlet. During the Holocene glaciers C93 and C79 retreated up-valley and today they are located about 14 km from the coast (Inland Waters Branch, 1969). Following glacial retreat, the valley became partially submerged beneath the sea between 11,335 cal yr BP to 6020 cal yr BP according to Allard (1996). Alternating layers of peat and aeolian sands and silts (~2-3 m) cover the glaciofluvial terraces of the valley, where an extensive network of syngenetic ice-wedge polygons 30 have developed after 6000 cal yr BP (Fortier and Allard, 2004). Mounds of reworked till and ice-contact stratified sediments mark a former position of the glacier front in the valley.





Bylot Island belongs to the Arctic Cordillera and the Northern Arctic terrestrial ecozones. The MAAT between 1971 and 2000 in Pond Inlet (NU, Canada) was -15.1 ± 5.1 ˚C, increasing slightly to -14.6 ± 4.9 ˚C between the 1981 and 2010 (Environment Canada, 2015). No significant trends in precipitation have been observed over the last decades, with a mean annual precipitation of 189 mm, much of it falling as snow. Thawing and freezing indices averaged (1981-2010) 473 degree-

days above 0˚C and 5736.1 degree-days below 0˚C, respectively (Environment Canada, 2015). Vegetation in the valley is typical of Arctic tundra environments and is largely determined by soil moisture and slope (Duclos et al., 2006). Wetlands occur in low-lying areas, commonly with low-centred polygons, and are typically dominated by grasses and sedges. Mesic tundra is found in areas characterized by better-drained soils (i.e. plateaus and hillslopes). The climatic and vegetation conditions determine the presence of continuous permafrost in the ice-free areas of Bylot Island. Locally, permafrost

thickness >400 m has been detected from thermal measurements on nearby Somerset and Devon Islands (Smith and Burgess, 2002). Active layer thickness varies from ~ 1-m in drained unvegetated sands and gravels, to ~ 0.3-0.7 m in peaty and silty soils (Allard et al., 2016).

**3 Methods**

A large body of massive ice was found exposed beneath ~ 1.7 m of sediments along the headwall of a thaw slump in the

Qarlikturvik Valley, ~10 km from the terminus of C93 and C79 glaciers (Fig. 1). Since a great amount of slump material covered the ice, the exposure was cleaned and enlarged to allow a better description and sampling. The excavated section along the headwall of the slump attained a height exceeding 7 m and extended laterally over 10 m. The lower and lateral contacts of the massive ice have not been reached. The exposure was subdivided into three units, from bottom: A) Massive ice; B) Sand and gravel and C) Muddy sand diamicton overlaid by peat (Fig. 2). The thaw depths were measured with a steel

probe at every 10 metres along a 150-metre transect that started at the upper headwall of the thaw slumps. Samples from the massive ice (unit A) were collected using an axe and a portable core-drill equipped with an 8 cm diamond carbide core barrel. Ice cores were extracted every 10 cm or less from depths ranging from one to three metres below the surface (Fig. 2) without reaching the bottom of the ice body. For comparison, modern glacier ice (englacial ice) was sampled from freshly collapsed ice blocks at the margin of glacier C93 located a few kilometres up the valley (Fig. 1). Wedge ice, segregated ice

and snow, were also sampled for geochemical analysis at the site located nearby (< 1 km) the massive ice exposure. Samples (n=5) from the sediments (units B and C) overlying the massive ice were collected and characterized. All samples (n=80) were melted in the field in sealed polyethylene bottles, shipped and stored in our laboratory for further analysis.

A cryostratigraphic approach was used to describe the massive-ice body and the overlying sediments (Fortier et al., 2012;

French and Shur, 2010; Gilbert et al., 2016; Murton and French, 1994; Stephani et al., 2010). Cryostratigraphic units were delineated based on cryostructures and cryofacies. Cryofacies are bodies of frozen sediment that are visually distinct from adjacent frozen sediments based on their cryostructures, volumetric ice content, ice-crystal size and sediment texture.



Cryostructures refer to the shape, amount and distribution of ice within the frozen sediment. Gas inclusions visible within the ice and the deformation structures in the ice and sediments were also described (Hambrey and Lawson, 2000). To further investigate the cryostratigraphic characteristics of the massive ice body, all samples were observed under X-ray computed tomography (CT) scanning (*Siemens SOMATOM Sensation 64*), a technique that allows visualizing and reconstructing the

internal structure (2D and 3D) of permafrost samples at <1 mm resolution (Calmels et al., 2010; Dillon et al., 2008; Fortier et al., 2012). The complete CT scanning procedure used in this study is presented in supplementary material. Crystallographic analysis of the massive ice and modern glacier ice was conducted to describe their crystal size and shape (surface area, long-axis and circularity ratio) as these parameters contain information about the conditions under which the ice was formed (French and Shur, 2010). The crystalline structure was investigated through thin sections of ice mounted on a glass slide

under cross-polarized light. Thin sections were made by cutting the ice sample vertically and/or horizontally into thin slices (thickness: 0.2 to 0.4 mm) using the procedure outlined by Langway (1958). Measurements of c-axis orientations of the crystals were not possible since the horizontal orientation of the ice samples could not be ascertained. Fiji image analysis software was used to measure the area, long axis, circularity ratio of each crystal (Schindelin et al., 2012). Differences in the crystal shape (area, long axis, circularity ratio) of horizontal and vertical thin sections were tested with the Mann-Whitney U

test using R, which is a programming language and free software for data analysis and graphics (R Development Core Team, 2016).

The massive ice body, along with glacier C93 ice, ice wedges, intrasedimental ice and snow, were analyzed for their geochemical and isotopic ($\delta^{18}$O, $\delta$D) composition, an approach that can shed light into the origin of ground ice (Fujino and

Kato, 1978; Lacelle and Vasil'chuk, 2013). Sampling of the massive ice body was done at 10 cm vertical intervals or less, depending on the unit change; prior to sampling, at least 10 cm of the ice surface was removed with an ice axe. All samples were melted and filtered (0.45 μm diameter filter) prior to analyses. The concentration of major cations in the ice and snow ($Al_{tot}$, B, $Ca^{2+}$, $Fe_{tot}$, $K^+$, Mg, $Mn_{tot}$, $Na^+$, P, S, Si and Sr) was determined by inductively coupled plasma optical emission spectrometry (Vista Pro ICP-OES) at the University of Ottawa. Solutes are expressed in milligrams per litre and analytical

reproducibility was ± 1%. The stable isotope ratios of oxygen ($^{18}$O/$^{16}$O) and hydrogen (D/H) were determined using a Los Gatos Research high-precision liquid water analyzer coupled to a CTC LC-PAL autosampler. The results are presented using the $\delta$-notation ($\delta^{18}$O and $\delta$D), where $\delta$ represents the parts per thousand differences for $^{18}$O/$^{16}$O or D/H in a sample with respect to Vienna Standard Mean Ocean Water (VSMOW). Analytical reproducibility for $\delta^{18}$O and $\delta$D was ± 0.3‰ and ± 1‰, respectively.

The origin of the sediment overlying the massive ice was inferred from particle-size distribution, the clasts and sand-size quartz grain morphoscopy, and the geochemical and isotopic ($\delta$D-$\delta^{18}$O) composition of pore water. Particle-size distributions were determined by dry sieving at ½ φ intervals (size ranges -12 to 4 φ). The hydrometer method was used to determine the distribution of the finer particles smaller than 4 φ (ASTM Standard D422, 2007). Descriptive statistics (mean grain size,





sorting, skewness) and *Folk & Ward* sediment classes were determined using the RYSGRAN package for R (Gilbert et al., 2014; R Development Core Team, 2016). Fifty *in situ* pebble- to cobble-sized clasts were randomly collected from the uppermost unit (Unit C). These clasts were analyzed for shape, roundness, and lithology using techniques described by Benn (2004). Morphoscopic analyses of small quartz grains (0.5-1.0 mm) were conducted using a binocular microscope (Cailleux

and Tricart, 1963). Additionally, the pore ice in the sediments was analyzed for soluble ions (major cations) and δD-δ$^{18}$O composition following the method described above. A fragment of poorly decomposed peat sampled in unit B was radiocarbon dated by accelerator mass spectroscopy (AMS) (ULA-6505, Centre for Northern Studies, Université Laval). Radiocarbon age was calculated as –8033ln(F$^{14}$C) and reported in $^{14}$C yr BP (BP=AD 1950) (Stuiver and Polach, 1977) and then corrected to calendar years (cal yr BP) using Calib 7.10 and the IntCal13 calibration curve (Reimer et al., 2013; Stuiver

et al., 2017).

## 4 Results

### 4.1 Cryostratigraphy and properties of the massive ice body.

The exposed massive ice body (unit A) was > 10 m thick and had a clear to milky white appearance due to its high bubble content (Fig. 3a). With VWC near 100%, it refers to the "pure ice" cryofacies described by Murton and French (1994).

Occasional thin bands of sediments (sands and gravels) with suspended and crustal cryostructures were cutting across the ice (Fig. 3b). These discrete and planar bands were < 2-cm thick, sub-parallel one to the other and showed a dip direction (21˚ to 31˚) downward the ice body in the southeast direction. Crystallographic analysis of thin sections of the massive ice under cross-polarized light showed that the crystals had mostly bluish colours, suggesting that the ice crystals had similar orientations (Fig. 4 and 5a). Coarse-grained ice crystals characterized the massive ice body: long axis average of 7.97 ± 2.97

mm$^2$ (3.13-16.58 mm); average surface area of 34.9 ± 25 mm$^2$ (5.8 and 153.5 mm$^2$); average circularity ratio of 0.65 ± 0.09, indicating the crystals were relatively rounded (Fig. 6). No significant differences in the shape properties (surface area, long axis, circularity ratio) were observed between the horizontal and vertical thin sections (Mann-Whitney-Wilcoxon test, p>0.05), indicating that the ice crystals were nearly equiaxial. The glacier C-93 ice displayed ice crystals with varying colours (Fig. 4). The ice crystals were larger (average surface area of 125.29 ± 148 mm$^2$) than those of the massive ice body

(Fig. 4 and 6). Glacier C-93 ice consists of relatively rounded crystals as their mean circularity ratio averaged 0.65 ± 0.01. The total volumetric content of gas inclusions varied from 2 to 10 % for both the massive ice and glacier C-93 ice, with bubbles being mostly located along grain boundaries (Fig. 5a). The massive ice body contained three types of gas inclusions: A) spherical bubbles; B) flattened disks and C) clusters of deformed and coalescent bubbles (Fig. 5b, c, d). The tiny gas bubbles had a mean surface area of 0.85 ± 1.04 mm$^2$ with circularity ratio averaging 0.89 ± 0.17. Gas bubbles observed in

glacier C-93 were mostly spherical and small, with a few clusters of deformed coalescent bubbles. The gas bubbles had an average circularity ratio of 0.89 ± 0.18 and a mean surface area of 0.13 ± 0.41 mm$^2$, respectively.



The dominant cations in the massive ice body were $Ca^{2+}$, $Na^+$ $Mg^{2+}$, $K^+$ and S; all with low abundances (< 1.76 mg/L; Fig. 7). The cation concentrations of glacier C-93 ice were very similar to those of the massive ice; whereas the ice wedge had slightly higher cation concentrations, with average concentrations ranging from 1.11 mg/L for $Mg^{2+}$, to 3.32 mg/L for $Ca^{2+}$.

The $\delta D$-$\delta^{18}O$ composition of the massive ice, along with those from glacier C93 and other types of ground ice present in the valley are shown in Fig. 8 (Fortier et al., 2018). The $\delta^{18}O$ composition of the massive ice had a narrow range (average $\delta^{18}O$: -34.0 ± 0.4‰) with D-excess ($d$=$\delta D$ - 8 $\delta^{18}O$) averaging 6.6 ± 2.5‰. The $\delta^{18}O$ composition of the massive ice was much lower than that of the snow (average $\delta^{18}O$: -30.4‰ ± 1.8), ice-wedge ice (average $\delta^{18}O$: -25.6‰ ± 0.95) and glacier C93 ice (average $\delta^{18}O$ -25.0 ± 3.1‰). The deuterium excess of snow, ice wedge, glacier C93 and intrasedimental ice samples averages 8.9 ± 3.4‰).; 9.3 ± 7.1‰, 5.2 ± 5.8‰ and -52.4 ± 31.4‰ respectively. In a $\delta D$-$\delta^{18}O$ diagram, the samples from the massive ice, glacier C-93, snow and ice wedges are distributed along linear regression slope values of 6.0 ($R^2$=0.44), 7.1 ($R^2$=0.95), 6.6 ($R^2$=0.96), and 5.2 ($R^2$=0.36), respectively.

**4.2 Cryostratigraphy and properties of the overlying sediments**

The massive ice body was covered by ~1.7 m of sediments. The thaw depths measured in late July (2013) ranges between 19 and 55 cm (mean: 30 cm ± 9 cm). A sharp, sub-planar and unconformable contact separated the ice from the overlying sediments (units B and C) along the exposed section (Fig 2). Unit B (~115 cm thick) directly overlies the massive ice body and has an ice-poor sediment cryofacies with a structureless cryostructure, essentially made of pore ice. Unit B can be subdivided into three sub-units: $B_1$: Coarse sandy gravel, $B_2$: Sandy gravel, $B_3$: Stratified gravelly sand. All sub-units were
texturally similar, consisting mostly of sands and gravels, with silt- and clay-sized particles constituting < 3% of the sediment (Fig. 9 and 10). The grain size fraction > 32 mm was not included in profile distribution, but abundant cobbles were randomly observed within the bottom sub-units. Sediments are poorly sorted to very poorly sorted (sorting values ranging between 3 and 5) and show a general fining upward trend (Figs. 9 and 10). Sub-units $B_1$ and $B_2$ are both characterized by unstratified sandy gravel that becomes finer towards the uppermost sub-unit $B_3$, which consists of stratified
gravelly sand with thin sub-horizontal laminae (< 1 cm). Morphoscopic analysis of sub-unit $B_3$ shows that quartz grains were mostly angular (70%) to sub-angular (smooth and polished, glazing grains: 25%). Rounded and frosted grains represent < 5 % of the total sand fraction. Plants fragments located at the base of sub-unit $B_3$ were dated to 786 cal yr BP (1σ range: 746-883). The sand and gravel sequence of Unit B is sharply overlain by ~ 55 cm of an unstratified (massive) thawed diamicton, with abundant pebble- to cobble-sized clasts (Unit C; Fig. 2 and 9). The matrix (sandy silt) was very poorly sorted and the
grain size distribution tended to be fine-skewed. Clasts had a wide range of shapes, with predominance in the sub-angular and sub-rounded classes (76%). Only about 24% of all clasts were rounded or angular. Gneiss was the dominant cobble type





with an average of 60 % in the matrix whereas sedimentary and igneous rocks average 36 % and 4 % respectively. A thin, continuous and irregular layer of dark fibrous peat with roots (sub-unit C2) overlaid the diamicton.

Supernatant water samples (n=5) from the ice-poor sediments in Unit B showed markedly higher $\delta^{18}O$ and cation values

compared to the underlying massive-ice body (Figs. 7 and 8a). In Unit B, the $\delta^{18}O$ values become progressively higher, from -25.7‰ to -15.4‰, as we move upward from the ice-sediment contact towards the surface (Fig. 8a). The cations content in the ice-poor sediment is 1 to 2 orders of magnitude higher than in the massive ice and also shifts to higher concentrations towards the surface (Fig. 7).

## 5 Discussion

The cryostratigraphic and crystallographic properties of the massive ice along with its isotopic and geochemical composition indicate that the exposed ice consists of relict Pleistocene englacial ice buried and preserved in the permafrost of Bylot Island. We first discuss the origin of the massive ice then the geomorphic processes that lead to the burial of the ice and preservation. Table 1 summarizes the cryostratigraphic and crystallographic properties of different types of tabular massive ice described in the literature: buried glacier ice (basal and englacial), massive segregated-intrusive ice, and buried

snowbanks.

### 5.1 Origin of the massive ice

The appearance and structure of buried massive ice are similar to those of englacial ice typically observed at the margin of glaciers, ice caps or ice sheets. The buried massive ice body has a whitish appearance owing to its high concentration of gas inclusions. Coarse-bubbly ice is the most abundant type (90-95%) of englacial ice found in glaciers (Allen et al., 1960). Our

results also show that the cross-sectional area of the crystals of the buried massive ice is smaller than that of neighbouring C93 glacier ice, but there is no significant difference in their circularity ratio (Mann-Whitney-Wilcoxon test, p=0.89). However, the difference in ice crystal size is not unforeseen since the ones from glacier ice can show variations on the order of a few millimetres to several centimetres in diameter (Gow, 1963; Romanovsky and Cailleux, 1970; Svensson et al., 2003; Thorsteinsson et al., 1997). Patterns of preferred crystal orientation combined with the occurrence of deformation features in

the form of debris bands suggest that the ice has been subjected to long-continued shear stress caused by the motion of the glacier (Knight, 2013; Lawson, 1979). The debris bands cross-cutting the buried glacier ice are comparable to those observed in a zone of the terminus of Stagnation Glacier on Bylot Island, where basal sediments were transported to the glacier surface through shear planes (Moorman and Michel, 2000).

Cations and stable water isotopes measured in the buried massive ice also support its glacial origin. The low cation content in the buried massive ice is statistically similar to that of the ice of glacier C93. Although a slope of 6.0 was calculated between $\delta D$-$\delta^{18}O$ of the buried ice, lower than the GMWL (slope =8; Craig, 1961), this was due to the small range of the





data ($\delta^{18}O$: -34.4 ‰ to -33.4 ‰) which prevented the calculation of a reliable regression slope (Lacelle et al., 2007). The $\delta D$-$\delta^{18}O$ values of the buried massive ice are similar to those of Pleistocene-age ice from the Barnes Ice Cap (Zdanowicz et al., 2002). This is supported by the average D-excess of 6.6 ‰ ± 2.5 for the buried massive ice, which is within the range of the values of Barnes Ice Cap and glacier C-93 (5.2 ‰ ± 5.8) on Bylot Island.

The average $\delta^{18}O$ value of the buried massive ice (-34.0 ‰ ± 0.35) is much lower than that of the other ice types sampled in the study area (Fig. 8b). The $\delta^{18}O$ values of the buried massive ice are also lower than in the ice of Penny Ice Cap from the last glacial period ($\delta^{18}O \sim$ -31.3 ± 1.1‰; Fisher et al., 1998), but more alike the late Pleistocene ice of Barnes Ice Cap (Zdanowicz et al., 2002). The late Pleistocene $\delta^{18}O$ values on Barnes Ice Cap were 6-10‰ lower than the expected $\delta^{18}O$

[modern - last glacial maximum] and interpreted to be the product of two factors: 1) temperature effect: cold climate during the last glacial period that caused a $\delta^{18}O$ shift of 6-7‰ at that latitude (Fisher and Alt, 1985); 2) elevation effect: ice sourced from higher elevation on Foxe Dome (~ 2200-2400 m) that caused an additional depletion in $\delta^{18}O$ of local precipitation (Hooke and Clausen, 1982; Zdanowicz et al., 2002). Given that the $\delta^{18}O$ values of the buried massive ice is ~10.5‰ lower than the amount weighted $\delta^{18}O$ of precipitation at Pond Inlet (–23.8‰; IAEA/WMO, 2016), and that the cooler climate

during the LGM accounts for 6-7‰ shift at the latitude of Bylot Island, the remaining 3.5-4.5‰ difference suggests that the ice was sourced from higher elevation.  Klassen (1993) suggested that the alpine glaciers on Bylot Island were larger but did not change much in elevation during the late Pleistocene. Mega-scale glacial lineations and streamlined landforms were mapped on the floor of Navy Board Inlet, Eclipse Sound and Lancaster Sound by De Angelis and Kleman (2007), and interpreted as a product of LIS ice streams. We proposed that the study area was most likely an area of confluence of LIS ice

and local alpine glaciers during the LGM as an ice stream moved through Navy Board Inlet and onto south-western Bylot as proposed by Dyke and Hooper (2001).

**5.2 Burial and preservation of late Pleistocene glacier ice in permafrost**

The texture (sand and gravel), stratification, and poorly sorted nature of the sediments (unit B) directly overlying the buried

glacier ice suggest a glaciofluvial ice-contact deposit (Boulton, 1972). Similar sediment characteristics were obtained by Fortier and Allard (2004) for glaciofluvial sands (angular grains: 75%) located a few kilometers away from the study site. The abundance of angular grains (70%) in unit B indicates little abrasion, which is consistent with a transport by glaciofluvial water where grains were carried over short distances allowing little wear. Furthermore, the sharp and unconformable contact between the buried glacier ice and the overlying sediment suggests that thermal erosion caused by

sediment-laden waterflow affected the top of the ice of unit A. The uppermost unit (C) is a diamicton, which has undergone reworking by non-glacial processes such as gravitational mass-wasting (e.g. debris flow and solifluction). In ice-contact environments, the sediment cover is subject to several cycles of subsidence and redeposition as the ice undergoes progressive and partial melting (Schomacker, 2008). The shape of the clasts found within this unit provides evidence that it has





experienced active glacial transport, which is also supported by the occurrence of erratic clasts (gneiss) derived from distal bedrock located several kilometres from the study site. To summarize, large stagnant ice blocks would have covered by glacigenic sediments accumulations at or near the ice margins during the advance and stagnation of a glacier within the Qarlikturvik Valley. The burial of the ice occurred as meltwater streams deposited sediments in direct contact with glacier

ice followed by the reworking and redeposition of supraglacial sediments, which formed a surficial cover of mud and sand, later affected by cryoturbations as is indicated by the incorporation of organic material dated at 786 cal yr BP (1σ range: 746-883).

A clear discontinuity in the $\delta^{18}O$ and cations profiles is observed at the ice-sediment contact and in the overlying units (Figs. 7 and 8). Strong contrasting profiles between buried ice and the overlying sediment are related to different sources of water

and freezing history. The supernatant water from the ground ice in the overlying sediments (units B and C) has much higher average $\delta^{18}O$ values of -17.6 ‰ ± 5.4. These values are comparable to $\delta^{18}O$ values obtained from precipitation at Pond Inlet between 1990 and 1992 (–23.8 ‰; IAEA/WMO, 2016) and modern segregation ice which generally has $\delta^{18}O$ values of -18 to -22 ‰ (Michel, 2011). The average concentrations of Ca, Na Mg and K in units B and C are 10-86 times greater than those in the buried glacier ice. The substantial increase observed in the cation content from the sediment layers can be

attributed to the great amount of mineral dissolution before the freezing of the water (Lacelle and Vasil'chuk, 2013).

Large blocks of buried late Pleistocene glacier ice were left undisturbed for several millennia owing to cold and dry climatic conditions that favored permafrost aggradation following deglaciation. The deposition of a cover of coarse and well-drained sediment exceeding the average active layer thickness of the area has probably been the most important factor limiting the melting of the ice. Furthermore, plant colonization and the development of a continuous vegetation cover with organic

accumulation have insulated the buried ice by reducing heat flow from the atmosphere to the permafrost during the summer and favoring heat loss during winter (French, 2010). The preservation potential of buried glacier ice on a millennial time-scale following the glacial retreat depends on the complex interactions between climate, geomorphology, and the physical properties of the sediment cover. Between 1999 and 2016, ground thermal regime monitoring in an intact low-centered polygon nearby the study site showed maximum active layer depths varying between 0.3 and 0.7 m while the buried ice is

located >1 m below the ground surface. While the properties of the sediment cover had positive feedback on the long-term preservation of the buried ice, thermal erosion and subsequent thaw slump activity were fundamental drivers of its degradation by exposing the ice and accelerating its melting (Kokelj et al., 2015; Lacelle et al., 2010; Segal et al., 2016).

## 6 Conclusion

This study confirms that the permafrost of Bylot Island contains remnants of Pleistocene glacier ice that survived the last

deglaciation. Geomorphic and cryostratigraphic observations along with the crystallographic properties of the massive ice




suggest its englacial origin. Evidence in support of the englacial origin of the massive ice are the following: 1) Sharp and unconformable upper contact between the ice and the overlying glaciofluvial sediments; 2) Clear to whitish ice, with large crystals; 3) Bubble-rich ice, with small gas inclusions mainly located at crystal junctions; 4) Occasional debris bands of sand and fine gravel cross-cutting older debris-free ice; 5) Geochemical similarities with contemporary glacier ice. The

geochemical data show a clear discontinuity at the buried ice-sediment contact as well as the very low cation content similar to that of modern glacier ice. An origin of massive ice from a Pleistocene glaciation is suggested based on the isotope data: the buried englacial ice isotopic composition is highly depleted in heavy isotopes, similarly to regional Pleistocene glacier ice. Glacier-derived permafrost contains ice that predates the aggradation of the permafrost and represents unique environmental archives to reconstruct paleo-environmental conditions at the study site. Although stable isotopes do not yield

information on absolute ages, these data show that the glacier ice body was buried and preserved in the permafrost of Bylot Island for thousands of years. The sedimentological data and interpretations presented in this study demonstrate that the first phase in the burial of the ice involved glaciofluvial deposition directly on the ice, which was followed by mass wasting. Knowledge regarding the occurrence, origin, and preservation of buried glacier ice is of a great interest due to its potential impacts on the landscape stability upon melting. Since Bylot Island has experienced several periods of native and foreign

Pleistocene glaciations and based on our findings, we propose that buried glacier ice is widespread on the island. In a context of climate change, active layer deepening and increased activity of slope processes, such as active layer detachment slides, thaw slumping and thermo-erosional gullying, will very likely expose buried glacier ice and initiate major landscape changes, with cascade effects on the ecosystems.

### 7 Acknowledgements

We are grateful to E. Godin, K. Larrivée, N. Perreault, A. Veillette and S. Veuille, for their help in the field and in the laboratory. We also thank the team of Prof. G. Gauthier (U. Laval) and the staff of the Sirmilik National Park for logistical support and access to Bylot Island. This project was funded by ArcticNet, the Natural Sciences and Engineering Research Council of Canada (NSERC), the Polar Continental Shelf Program (PCSP) of Natural Resources Canada, the Northern Scientific Training Program (NSTP) of the Canadian Polar Commission, the NSERC Discovery Frontiers grant 'Arctic

Development and Adaptation to Permafrost in Transition' (ADAPT).





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

30



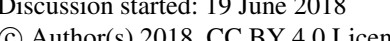


**Figure 1: Location of the study area within the Qarlikturvik Valley (valley of glaciers C79 and C93), Bylot Island, Nunavut, Canada. The red square shows the location of the massive ice exposure. Wedge ice, segregated ice and snow were also sampled within the area delineated by the red square. The yellow star indicates the sampling location of C93 glacier ice. The thaw depths**
5 **were measured with a steel probe at every 10 metres along a 150-metre transect.**

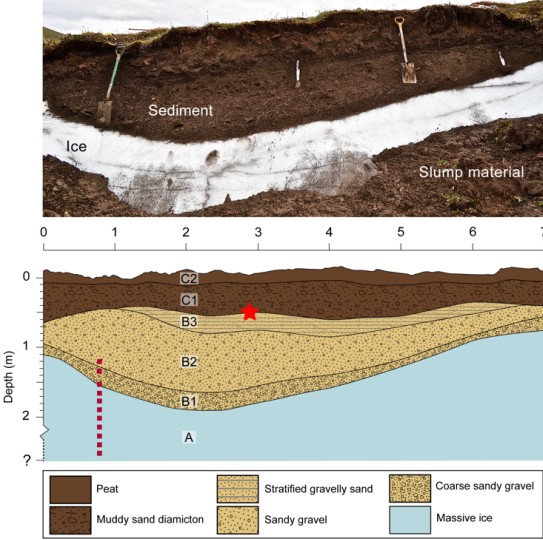

**Figure 2: A photograph and a schematic cross section showing generalized stratigraphy of the massive ice exposure and the overlying sediments. The lower and lateral contacts of the massive ice have not been reached. The thaw depths measured in late**
10 **July (2013) at the headwall reaches 55 cm.**



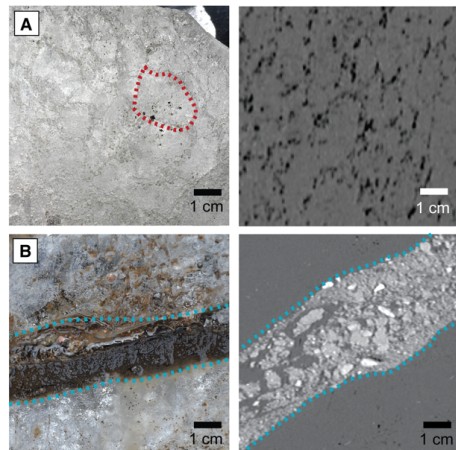

**Figure 3: Photographs (left) and CT scans (right) of the massive ice body. A) Pure ice facies. The red-dotted line highlights one single crystal; B) Ice-poor sediment (sands and gravels) with suspended and crustal cryostructures forming a band in the massive ice.**

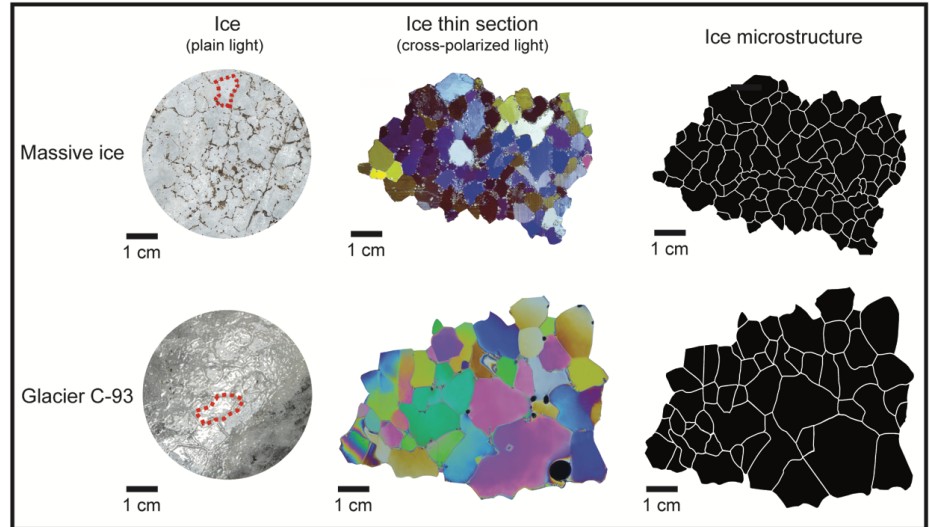

**Figure 4: Comparison between the massive ice body and modern glacier ice (glacier C93, Bylot Island). The first column shows unprocessed photographs of the ice taken under plain light with surficial sediment inclusions highlighting the crystal boundaries. The red-dotted line highlights one crystal. The second column shows thin sections of ice sample viewed under direct cross-polarized light. The third column shows the microstructure (crystal boundaries) extracted from the thin section photograph.**



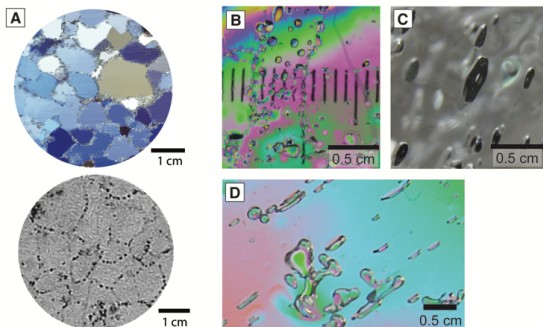

**Figure 5: A) A thin section of the massive ice viewed under cross-polarized light and a transverse cross-section from a scan showing the gas inclusions within the ice. (Air = black; Ice = dark grey). Photos to the right show patterns of gas inclusions; B)**
5 **Small (sub-mm to mm) spherical bubbles (vertical bars are from measuring ruler of microscope stage); C) Small disks up to 6 mm in diameter; D) Coalescent bubbles and small disks all flattened in the same direction.**

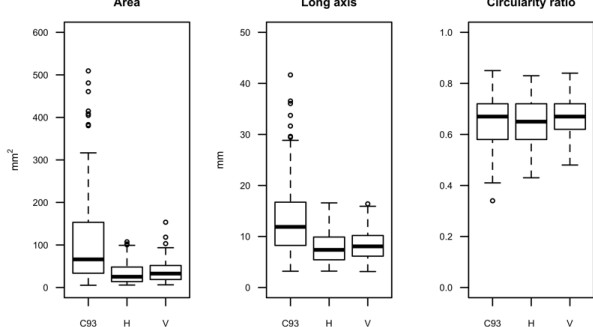

**Figure 6: Boxplots comparing the distribution of ice crystal characteristics (area, long axis, circularity ratio) of horizontal thin-**
10 **sections (H) and vertical thin-section (V) obtained from massive ice samples. C93 represents data obtained from a sample of modern glacier ice sampled from glacier C93.**



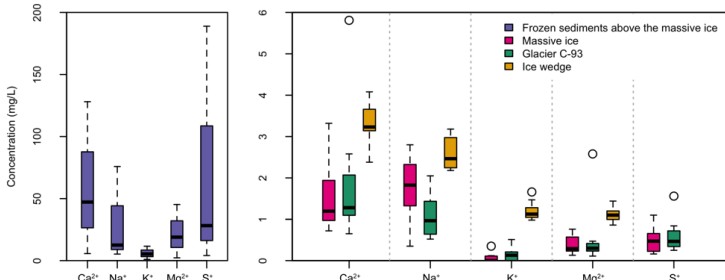

**Figure 7: Boxplots showing soluble cation concentration of the massive ice, ice wedge, glacier C-93 and intrasedimental ice sampled within the sediments layers covering the massive ice unit.**

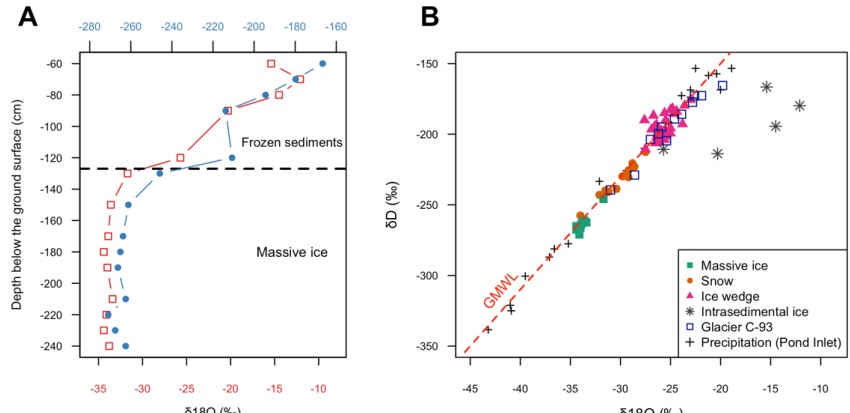

**Figure 8: A) Stable O-H isotopic depth profile including both the massive ice unit and the intrasedimental ice from the sediment cover; B) δ$^{18}$O-δD diagram of the massive ice and other types of ground ice (ice wedge, intrasedimental ice), snow and modern glacier ice (C-93) sampled on Bylot Island. The red-dashed line represents the Global Meteoric Water Line (GMWL): d=δD–8 δ$^{18}$O+10 (Dansgaard, 1964). Also shown are the δ$^{18}$O and δD values of precipitation recorded at Pond Inlet between 1990 and 1992**
10 **(IAEA/WMO, 2016).**



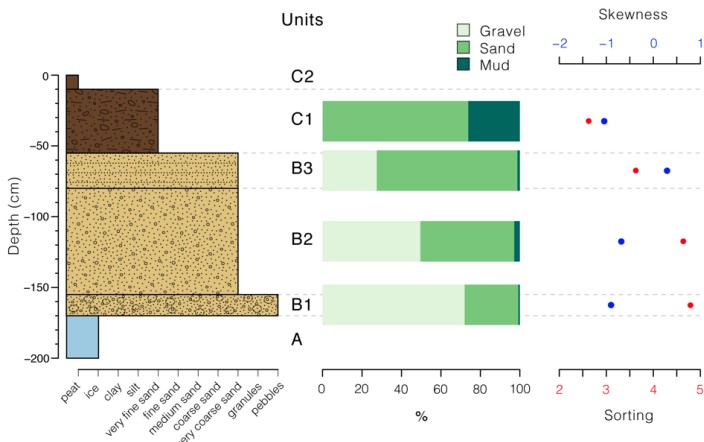

**Figure 9: Sedimentological data from the stratigraphic section. From left to right: A stratigraphic log showing the mean grain size of each unit, gravel-sand-mud percentages, skewness and sorting.**

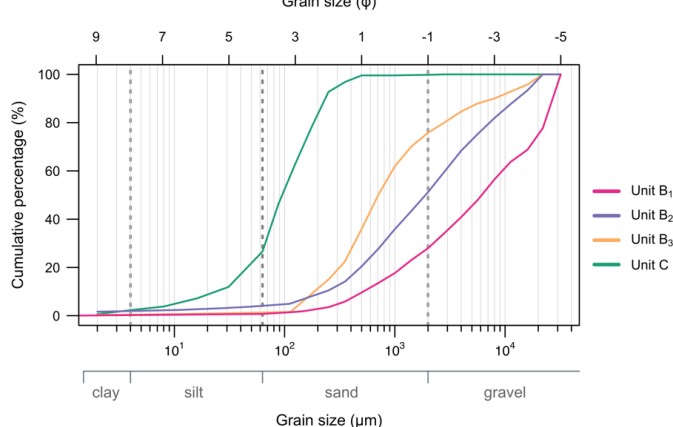

**Figure 10: Grain size distribution curves of the sedimentary units (two samples per sub-units). The gravel fraction was removed prior to analysis.**




| | Ice type | General appearance | Texture | Fabric | Gas inclusions | Ref. |
|---|---|---|---|---|---|---|
| Intrasedimental ice | **Massive segregated-intrusive ice** | • Clear<br>• Sediment-free to sediment-poor | • Variable crystal size, but mostly medium to large grains (cm-scale)<br>• Slightly elongated crystals [1]<br>• Ahnedral to subhedral | • Preferred near-vertical oriented c-axis [1] | • Bubble-poor to bubble-rich<br>• Small (mm- to cm-scale)<br>• Elongated and tubular bubbles [1]<br>• Trains of small spherical bubbles [1]<br>• Inter- and intra-crystalline | 1, 2, 3, 4, 5, 6 |
| Buried surface ice | **Buried glacier ice (englacial)** | • Clear to milky white<br>• Deformation structures [2]<br>• Foliations of bubble-poor and bubble-rich ice | • Wide range of crystal sizes (sub-mm to tens of cm)<br>• Mostly large grains (cm-scale)<br>• Equigranular<br>• Interlocked crystal boundaries | • Random or preferred c-axis orientation [3] | • Bubble-poor to bubble-rich<br>• Located at crystal junctions<br>• Bubbles truncated at the ice-sediment contact | 1, 7, 8, 9, 10, 11 |
| | **Buried glacier ice (basal)** | • High sediment content<br>• Debris laminations<br>• Deformation structures [2]<br>• Suspended pebbles and cobbles | • Small grains (mm-scale)<br>• Ahnedral to subhedral | • Weak to strongly oriented c-axis | • Bubble-free to bubble-poor<br>• Small (μm to mm-scale)<br>• Flattened bubbles<br>• Bubbles truncated at the ice-sediment contact<br>• Located at crystal junctions | 11, 12, 13, 14 |
| | **Buried snowbank** | • Milky white to light brown<br>• Loosely compacted structure<br>• Bands of nearly horizontal pale brown ice<br>• Organic inclusions (e.g. twigs and leaf fragments) | • Small grains (mm-scale)<br>• Crystal size area usually reaches a few mm[2]<br>• Subhedral to euhedral<br>• Equigranular | • Random c-axis orientation | • Bubble-rich<br>• Layers of bubbles and dispersed in the ice<br>• Small spherical bubbles (μm- to mm-scale)<br>• Elongated and tubular bubbles<br>• Vertically oriented bubbles | 15, 16, 17 |

[1] Parallel to the heat flow direction, indicating that freezing is downward.
[2] Occurrences of debris bands, boudinage and pinch-and-swell structures, folding, thrust-faulting).
[3] Depending on the position of the ice within the glacier or ice sheet at the time it becomes buried.

**Table 1: Comparisons of the physical properties of different massive tabular ground ice found in the permafrost. References: (1)Pollard, 1990; (2)Yoshikawa, 1993; (3) Harry et al., 1988; (4) Gell, 1976; (5) Mackay and Dallimore, 1992; (6) Dallimore and Wolfe, 1988; (7) Rigsby, 1953; (8) Gow, 1963; (9) (Ingólfsson and Lokrantz, 2003); (10) (Tison and Hubbard, 2000);**
5 **(11) (Knight, 1997); (12)(Sharp et al., 1994); (13) (Sugden et al., 1995); (14) (Murton et al., 2005); (15) (Lacelle et al., 2009); (16)(Pollard and Dallimore, 1988); (17) (Petrenko and Withworth, 1999).**