# Peer review of "Origin, burial and preservation of late Pleistocene-age glacier ice in Arctic permafrost (Bylot Island, NU, Canada)"

_The Cryosphere, 2018_

## Referee Comment (RC1) · Anonymous Referee #1 · 16 Jul 2018

In this manuscript the authors present a description of buried massive ground ice at Bylot Island (Nunavut, Canadian Arctic) and results on permafrost cryostratigraphy, ice crystallography, stable water isotopes and cation contents to discuss the origin, burial and preservation of the studied massive ice body. Based on their results the authors conclude that the massive ground ice originates from Late Pleistocene glacier ice that has been buried with glacigenic sediments. A protective cover of sediments and peat ensured its preservation. The manuscript provides valuable information by adding a case study of buried glacier ice at Bylot Island based on new data. Results, discussion and conclusions are well supported by figures and references. The implications are relevant as quite some Arctic landscapes are determined by remnants of former glaciations preserved and hidden in permafrost that is vulnerable to climate warming. Melting of these massive bodies may have a significant impact on ground stability, landscape and ecosystem dynamics. On the other hand, buried glacier ice holds potential for paleoclimate reconstruction. All in all, the manuscript addresses a relevant research topic and contributes new and significant information. Hence, it is in general suitable for publication in The Cryosphere. However, there are some smaller issues that prevent me from considering the manuscript as publishable in its present form (see comments below). Hence, the manuscript needs some revision before it can be resubmitted for publication.

Specific comments:

P1L24: The statement on "most of the arctic landscapes..." does not hold when you consider the vast landmass of Beringia.

P2L1-7: Please check the references. Apparently, some references are mixed up (i.e. for Antarctica).

P2L30-34: The structure of the sentences can be improved (avoid the parenthesis).

P4L2: How far away is Pond Inlet? Can you mark it in Figure 1?

P4L27: Except for the massive ice samples I assume.

P6L14: Does VWC stand for volumetric water content? Please clarify.

P6L20: Should read mm for long axes.

P7L5-12: For a better overview I suggest to add a table to the manuscript providing the basic statistics for $\delta$18O, $\delta$D, d-excess (max, mean, min, sd), slope, intercept, number of samples for each type of ice/water.

P8L32: The slope of the C93 ice is below the GMWL, too. Are there any information

on past (ice cores?) and modern slopes (LMWL of IAEA stations?) available for your study region?

P9L2-4: Provide $\delta$18O numbers for Barnes ice cap for comparison. In the cited paper no d-excess values of Barnes ice cap are given, so it's not possible to compare your values.

P9L16: Can you provide estimations on the elevation difference for the ice source compared to today with respect to the 3.5 to 4.5‰ in $\delta$18O? Is there any indication of the age (i.e. more detailed than Late Pleistocene) of the studied buried glacier ice? Given the climate instability known from Greenland ice cores also abrupt climate changes may explain the additional 3.5 to 4.5‰ in $\delta$18O.

P14L12: Provide an URL for this dataset. Currently it isn't possible to find it.

Figure 1: It would be good to add an additional map (or enlarge the second provided map) of the entire Bylot Island to show the study site in the regional context of Bylot Island and the other sites mentioned the regional setting section (Lancaster Sound, Navy Board Inlet, Eclipse Sound).

Figure 2: What does the red star represent? Please clarify the meaning of the red dots in the left part (isotope and hydrochemistry samples?) and mark the position of the radiocarbon sample.

Figure 8a: It would be good to add d-excess to the figure (maybe replace $\delta$D by d-excess). Add the title for the upper x axis. $\delta$18O needs superscript.

---

## Referee Comment (RC2) · M. Fritz (Referee) · 10 Aug 2018

The manuscript provided by Coulombe et al. investigates the origin of massive ground ice in permafrost of Bylot Island (Nunavut, Canada). The authors argue that it is necessary to differentiate the origin of massive ground ice to model its spatial distribution and abundance for further landscape sensitivity analyses in times of permafrost thaw and landscape change. The manuscript presents detailed description of physical and geochemical properties of supposedly buried glacier ice. Field observations on cryostratigraphy and lithology are combined with laboratory analysis of grain-size distribution, ice crystallography, stable O-H isotopes and cation concentrations of the massive

ice and surrounding sediments including intrasedimental ice and ice wedges.

Based on own data and under consideration of existing literature on buried massive ground ice, the authors conclude that Bylot Island contains remnants of Pleistocene glacier ice that survived the last deglaciation. They suggest an englacial origin rather than a basal ice facies which is more common within the Wisconsin Arctic moraine belts. This conclusion is very likely to be true because contemporary glaciers are close and offer an excellent object to compare ice structures and englacial debris of relict and contemporary glacier ice. Finally, the authors discuss geomorphic processes that led to the burial and preservation of the ice.

The present study is of great interest to the Arctic research community in a context of recent warming, which is particularly strong in the high Arctic. Active layer deepening and increased activity of slope processes, (i.e. active layer detachments, thaw slumping and thermoerosion) expose such buried ice and will initiate landscape changes and associated effects on the ecosystems through lateral matter mobilization and surface disturbance.

The authors present original data and provide a thorough description of the methods. In general, this topic and the presented data are of interest for readers of The Cryosphere and especially for researchers studying permafrost and especially ground ice as environmental archive. The language is generally good and the figures and tables, in most cases, usefully complement the text.

There are some points that prevent the manuscript to be published as it is. I suggest the manuscript to be accepted after minor revisions.

**General comments:**

Please provide good arguments why you have measured major cations only and not anions? Both would be necessary to get a comprehensive understanding of the ion composition and water origin. What about standard parameters such as electrical conductivity and pH measurements?

The results are sometimes written in past tense and sometimes in present tense. Especially in 4.1 they are in past tense throughout and suddenly in 4.2 present tense pops up. Make sure you use one tense throughout.

The manuscript is rather short, which I personally like, but it contains more than 100 references although it is clearly not a review paper. The authors should find a way to consolidate and shorten the reference list a bit.

Sedimentological data is provided in figures 9 and 10. Since Figure 9 already provides information on gravel-sand-mud percentages and on skewness and sorting, Figure 10 does not add a lot of new information and can be removed. This would lead to a better balance of text vs. number of figures/tables.

All the original measurement data on stable isotopes, cation concentration, grain-size properties and crystallographic data as well as the calculated parameters such as slope, D-excess etc. should go into a table into the supplement of the paper or archived in PANGAEA before final publication of the manuscript.

**Specific comments:**

For specific comments see also the annotated and attached pdf-file.

*Michael Fritz (Alfred Wegener Institute, Helmholtz Centre for Polar and Marine Research)*

Please also note the supplement to this comment:
https://www.the-cryosphere-discuss.net/tc-2018-114/tc-2018-114-RC2-supplement.pdf

**Supplement:**

[revised manuscript text omitted]

---

## Author Comment (AC1) · 18 Sep 2018

Comments by the two referees were very enlightening and their suggestions useful; we are grateful for their input. His/her careful reading of the manuscript and his/her good knowledge of the subject-matter allowed providing relevant suggestions and additions to the manuscript. We treat each point raised in detail and with great interest.

Note that the line numbers given in this response refer to the revised version of the manuscript in track changes mode.

Referee #1

[Figure]

General comments

Comment 1: Referee #1: P1L23 - The statement on "most of the arctic landscapes..." does not hold when you consider the vast landmass of Beringia. Authors: We agree and modified for: "As most of the glaciated arctic landscapes [...]".

Comment 2: Referee #1: P2L1-2- Please check the references. Apparently, some references are mixed up (i.e. for Antarctica). Authors: Modification made. Problem with the reference manager.

Comment 3: Referee #1: P2L25-27 - The structure of the sentences can be improved (avoid the parenthesis). Authors: Modification made.

Comment 4: Referee #1: P3L11 - How far away is Pond Inlet? Can you mark it in Figure 1? Authors: We agree and location point has been added to the map (figure 1). In the Regional Setting section (p. 3, line 13), we also added that the study site is located "... at about 80 km north-west of the community of Mittimatalik (Pond Inlet)".

Comment 5: Referee #1: P4L17-18: Except for the massive ice samples I assume. Authors: Part of the massive ice samples were also melted.

Comment 6: Referee #1: P6L4 - Does VWC stand for volumetric water content? Please clarify. Authors: We modified for "With a volumetric ice content ...".

Comment 7: Referee #1: P6L10 - Should read mm for long axes. Authors: Modification made.

Comment 8: Referee #1: P6L27-31: For a better overview, I suggest to add a table to the manuscript providing the basic statistics for $\delta$18O, $\delta$D, d-excess (max, mean, min, sd), slope, intercept, number of samples for each type of ice/water. Authors: The data is provided on NordicanaD and we prefer to keep it as it stands.

Comment 9: Referee #1: P8L20: The slope of the C93 ice is below the GMWL, too. Are there any information on past (ice cores?) and modern slopes (LMWL of IAEA

stations?) available for your study region? Authors: The closest station of the of IAEA network is Pond Inlet and it only has data for two years, from January 1990 and December 1992. In section 5.2 of the Discussion, the amount weighted mean 18O for these two years is provided. These data are also plotted in figure 8b. We also added data and slope for Resolute Bay (n=59; 5 years) as the Pond Inlet has a rather small of data (n=20, 2 years), which prevents the calculation of reliable slope (LMWL). In Section 5.1 (Discussion), we compare our data to those obtained from cores sampled on the Barnes and Penny Ice Caps (p.8, lines 23-32).

Comment 10: Referee #1: P8L25: Provide $\delta$18O numbers for Barnes ice cap for comparison. In the cited paper no d-excess values of Barnes ice cap are given, so it's not possible to compare your values. Authors: The D-excess values of Barnes Ice Cap were provided by C. Zdanowicz and recently published in Lacelle et al., 2018 (Scientific Reports). We added citation to "Lacelle et al. 2018" to clarify this part.

Comment 11: Referee #1: P9L2-7: Can you provide estimations on the elevation difference for the ice source compared to today with respect to the 3.5 to 4.5‰ in $\delta$18O? Is there any indication of the age (i.e. more detailed than Late Pleistocene) of the studied buried glacier ice? Given the climate instability known from Greenland ice cores also abrupt climate changes may explain the additional 3.5 to 4.5‰ in $\delta$18O. Authors: The estimation on the elevation of the ice source were recently published in Lacelle et al., 2018 (Scientific Reports). There is no other indication of the age of the studied buried glacier ice. A fragment of poorly decomposed peat sampled in the overlying sediments was radiocarbon dated (p.9, 28-29). This surficial cover of mud and sand has been affected by cryoturbations as is indicated by the incorporation of this organic material. As regards to abrupt climate changes, we show a statistical argument for/against this in Lacelle et al., 2018 when we looked at variations in 18O for GISP2, Penny and Barnes Ice Caps. It is unlikely that we would have randomly sampled one of these short-lived 18O excursions.

Comment 12: Referee #1: P14L21: Provide an URL for this dataset. Currently it isn't

possible to find it. Authors: The dataset was being reviewed, but it is now available on NordicanaD with the DOI provided.

Comment 13: Referee #1: Figure 1 - It would be good to add an additional map (or enlarge the second provided map) of the entire Bylot Island to show the study site in the regional context of Bylot Island and the other sites mentioned the regional setting section (Lancaster Sound, Navy Board Inlet, Eclipse Sound). Authors: We agree. A general map of Bylot Island has been added to Figure 1.

Comment 14: Referee #1: Figure 2 - What does the red star represent? Please clarify the meaning of the red dots in the left part (isotope and hydrochemistry samples?) and mark the position of the radiocarbon sample. Authors: We agree. In the caption of figure 2, we added "The red star indicates the sampling location of the organic material and the red dots shows the sampling points for stable O-H isotope and hydrochemistry".

Comment 15: Referee #1: Figure 8a - It would be good to add d-excess to the figure (maybe replace $\delta$D by d- excess). Add the title for the upper x axis. $\delta$18O needs superscript. Authors: We agree, the figure 8a has been modified as suggested. We added the title for the upper axis and we replaced $\delta$D by the D-excess.

Please also note the supplement to this comment:
https://www.the-cryosphere-discuss.net/tc-2018-114/tc-2018-114-AC1-supplement.pdf

———————————————————

[Figure]

**Supplement:**

**Responses to the Reviewer's comments**

Comments by the two referees were very enlightening and their suggestions useful; we are grateful for their input. His/her careful reading of the manuscript and his/her good knowledge of the subject-matter allowed providing relevant suggestions and additions to the manuscript. We treat each point raised in detail and with great interest.

Note that the line numbers given in this response refer to the revised version of the manuscript in track changes mode.

**Referee #1**

**General comments**

Comment 1:
**Referee #1: P1L23 - The statement on "most of the arctic landscapes. . ." does not hold when you consider the vast landmass of Beringia**
*Authors:* We agree and modified for: "As most of the glaciated arctic landscapes […]".

Comment 2:
**Referee #1: P2L1-2- Please check the references. Apparently, some references are mixed up (i.e. for Antarctica).**
*Authors:* Modification made. Problem with the reference manager.

Comment 3:
**Referee #1: P2L25-27 - The structure of the sentences can be improved (avoid the parenthesis).**
*Authors:* Modification made.

Comment 4:
**Referee #1: P3L11 - How far away is Pond Inlet? Can you mark it in Figure 1?**
*Authors:* We agree and location point has been added to the map (figure 1).
In the Regional Setting section (p. 3, line 13), we also added that the study site is located "… *at about 80 km north-west of the community of Mittimatalik (Pond Inlet)*".

Comment 5:
**Referee #1: P4L17-18: Except for the massive ice samples I assume.**
*Authors:* Part of the massive ice samples were also melted.

Comment 6:

**Referee #1: P6L4 - Does VWC stand for volumetric water content? Please clarify.**

*Authors:* We modified for "*With a volumetric ice content …*".

Comment 7:

**Referee #1: P6L10 - Should read mm for long axes.**

*Authors:* Modification made.

Comment 8:

**Referee #1: P6L27-31: For a better overview, I suggest to add a table to the manuscript providing the basic statistics for δ18O, δD, d-excess (max, mean, min, sd), slope, intercept, number of samples for each type of ice/water.**

Authors: The data is provided on NordicanaD and we prefer to keep it as it stands.

Comment 9:

**Referee #1: P8L20: The slope of the C93 ice is below the GMWL, too. Are there any information on past (ice cores?) and modern slopes (LMWL of IAEA stations?) available for your study region?**

*Authors:* The closest station of the of IAEA network is Pond Inlet and it only has data for two years, from January 1990 and December 1992. In section 5.2 of the Discussion, the amount weighted mean $\delta^{18}O$ for these two years is provided. These data are also plotted in figure 8b. We also added data and slope for Resolute Bay (n=59; 5 years) as the Pond Inlet has a rather small of data (n=20, 2 years), which prevents the calculation of reliable slope (LMWL).

In Section 5.1 (Discussion), we compare our data to those obtained from cores sampled on the Barnes and Penny Ice Caps (p.8, lines 23-32).

Comment 10:

**Referee #1: P8L25: Provide δ18O numbers for Barnes ice cap for comparison. In the cited paper no d-excess values of Barnes ice cap are given, so it's not possible to compare your values.**

*Authors:* The D-excess values of Barnes Ice Cap were provided by C. Zdanowicz and recently published in Lacelle et al., 2018 (Scientific Reports). We added citation to "Lacelle et al. 2018" to clarify this part.

Comment 11:

**Referee #1: P9L2-7: Can you provide estimations on the elevation difference for the ice source compared to today with respect to the 3.5 to 4.5‰ in δ18O? Is there any indication of the age (i.e. more detailed than Late Pleistocene) of the studied buried glacier ice? Given the climate instability known from Greenland ice cores also abrupt climate changes may explain the additional 3.5 to 4.5‰ in δ18O.**

*Authors*: The estimation on the elevation of the ice source were recently published in Lacelle et al., 2018

(Scientific Reports). There is no other indication of the age of the studied buried glacier ice. A fragment of poorly decomposed peat sampled in the overlying sediments was radiocarbon dated (p.9, 28-29). This surficial cover of mud and sand has been affected by cryoturbations as is indicated by the incorporation of this organic material. As regards to abrupt climate changes, we show a statistical argument for/against this in Lacelle et al., 2018 when we looked at variations in 18O for GISP2, Penny and Barnes Ice Caps. It is unlikely that we would have randomly sampled one of these short-lived 18O excursions.

Comment 12:

**Referee #1: P14L21: Provide an URL for this dataset. Currently it isn't possible to find it.**

*Authors:* The dataset was being reviewed, but it is now available on NordicanaD with the DOI provided.

Comment 13:

**Referee #1: Figure 1 - It would be good to add an additional map (or enlarge the second provided map) of the entire Bylot Island to show the study site in the regional context of Bylot Island and the other sites mentioned the regional setting section (Lancaster Sound, Navy Board Inlet, Eclipse Sound).**

*Authors:* We agree. A general map of Bylot Island has been added to Figure 1.

Comment 14:

**Referee #1: Figure 2 -  What does the red star represent? Please clarify the meaning of the red dots in the left part (isotope and hydrochemistry samples?) and mark the position of the radiocarbon sample.**

*Authors*: We agree. In the caption of figure 2, we added "*The red star indicates the sampling location of the organic material and the red dots shows the sampling points for stable O-H isotope and hydrochemistry*".

Comment 15:

**Referee #1: Figure 8a - It would be good to add d-excess to the figure (maybe replace δD by d- excess). Add the title for the upper x axis. δ18O needs superscript.**

Authors: We agree, the figure 8a has been modified as suggested. We added the title for the upper axis and we replaced δD by the D-excess.

Referee #2

**General comment**

Comment 1:
**Referee #2: Provide good arguments why you have measured major cations only and not anions? Both would be necessary to get a comprehensive understanding of the ion composition and water origin. What about standard parameters such as electrical conductivity and pH measurements?**

The origin of the ice has already been established following the analysis of the physical properties (i.e crystallography), the isotopic composition (dD-18O, D-excess) and the low cation content. We did not need further information to infer its origin. We mainly used the cation content for comparison with glacier ice and other type of ground ice (i.e. interstitial ice, ice wedge). The cation content only showed that the low cation content in the buried massive ice is statistically similar to that of the ice of glacier C93 and had slightly lower cation concentrations that the ice wedge sampled nearby the buried ice exposure. The cation content also allowed showing a strong contrasting profile between the buried ice and the overlying sediment.

We have not measure the conductivity, as it is a proxy for total ion content. We already measured the cation content of the massive ice. As for pH measurement, pH data is meaningless for ground ice samples because of exchange of meltwater with atmospheric $CO_2$. The only way pH data would represent that of the ice is if the ice is melted in a glove box without a $CO_2$ atmosphere; hence we would not call this "standard parameters".

Comment 2:
**Referee #2: The results are sometimes written in past tense and sometimes in present tense. Especially in 4.1 they are in past tense throughout and suddenly in 4.2 present tense pops up. Make sure you use one tense throughout.**
Authors: We agree. This section has been reworked.

Comment 3:
**Referee #2: The manuscript is rather short, which I personally like, but it contains more than 100 references although it is clearly not a review paper. The authors should find a way to consolidate and shorten the reference list a bit.**
Authors: We agree. We shorten the reference list by removing 25 references.

Comment 4:
**Referee #2: Sedimentological data is provided in figures 9 and 10. Since Figure 9 already provides information on gravel-sand-mud percentages and on skewness and sorting, Figure 10 does not add a lot of new information and can be removed. This would lead to a better balance of text vs. number of figures/tables.**
Authors: We agree.

Comment 5:

**Referee #2: All the original measurement data on stable isotopes, cation concentration, grain-size properties and crystallographic data as well as the calculated parameters such as slope, D-excess etc. should go into a table into the supplement of the paper or archived in PANGAEA before final publication of the manuscript.**

Authors: The data is provided on NordicanaD.

**Specific comments**

Comment 1:

**Referee #2: Title- *"…Pleistocene-age glacier ice…"*. It would be easier to read the title (without changing the meaning) when you leave out the -age thing**.

*Authors:*  We prefer to keep the title as it stands.

Comment 2:

**Referee #2: P1L18 - What about anions?**

*Authors: Please see response to comment #1 in Reviewer #2-General comments.*

Comment 3:

**Referee #2: P1L23 - "As most of the Arctic landscapes…" - Better "some" because several millions of km² (Beringia) in the Arctic have not been glaciated throughout the Quaternary.**

*Authors: We agree* and modified for: "As most of the glaciated arctic landscapes […]"

Comment 4:

**Referee #2: P2L1-2 - Please check the references. Apparently, some references are mixed up (i.e. for Antarctica).**

*Authors:* Modification made. Problem the reference manager.

Comment 5:

**Referee #2: P2L25-27 – Something is wrong with this sentence**

*Authors:* Modification made.

Comment 6:

**Referee #2: P3L5 – Suggested revision: "*…the mountainous central section  forms a striking contrast…"***

*Authors:* Suggestion accepted

Comment 7:

**Referee #2: P3L19 – Clarification needed: *"submerged beneath the sea"***

*Authors:* We modified for "*Following glacial retreat, the valley became partially submerged […] as a result of a marine transgression*".

Comment 8:

**Referee #2: P4L4 – It would be better "Material and Methods" because you are also taking about the studied object and the samples that were taken.**

*Authors:* Modification made.

Comment 9:

**Referee #2: P4L17 – Clarification needed: *"melted"***

*Authors:* We modified for *"All samples (n=80) were thawed in the field [...]".*

Comment 10:

**Referee #2: P4L30 – Clarification needed: *"describe"***

*Authors:* We modified for *"[...] was conducted to measure their crystal size and shape [...]".*

Comment 11:

**Referee #2: P5L2-3 – Clarification needed: *"Measurements of c-axis orientations of the crystals were not possible since the horizontal orientation of the ice samples could not be ascertained."***

*Authors:* We modified for *"C-axis orientations of the crystals have not been measured since the horizontal orientation of the ice samples was not preserved following the sampling."*

Comment 12:

**Referee #2: P6L1 – This abbreviation was not mentioned before.**

*Authors:* We mentioned the three units (A, B, C) in Section 3 -Material and Methods (lines 13-14). It is mentioned that unit A refers to the massive ice body.

Comment 13:

**Referee #2: P6L2 – Clarification needed: "VWC".**

*Authors:* We modified for *"With a volumetric ice content [...]".*

Comment 14:

**Referee #2: P6L10 – Clarification needed: "mm$^2$".**

*Authors:* We modified for *"mm".*

Comment 15:

**Referee #2: P6L21 – Clarification needed*: "The gas bubbles had an average circularity ratio of 0.89 $\pm$ 0.18 and a mean surface area of 0.13 $\pm$ 0.41 mm$^2$ respectively."***

*Authors:* We removed "respectively*."*

Comment 16:

**Referee #2: P6L23 – Clarification needed: "The dominant cations in the massive ice body were Ca$^{2+}$, Na$^+$ Mg$^{2+}$, K$^+$ and S; all with low abundances (<1.76mg/L; Fig. 7)."**

*Authors:* We modified for *"Major cations in the massive ice body (i.e. Ca$^{2+}$, Na$^+$ Mg$^{2+}$, K$^+$ and S) all occurred in low concentrations (< 1.76 mg/L; Fig. 7)."*

Comment 17:

**Referee #2: P7L13 – Clarification needed *"[...] showed a general fining upward trend".***

*Authors:* We modified for *"were coarser at the base and finer near the top."*

Comment 18:

**Referee #2: P7L17 – Clarification needed: "[…]*were dated to 786 cal yr BP*".**

*Authors:* We modified for *"were dated to 885 ± 15 $^{14}$C yr BP (1164 cal yr BP; 1σ range: 1058-1204)."*

Comment 19:

**Referee #2: P8L1-6 – I suggest to incorporate the sentences of this introductory paragraph in the according sub-chapters 5.1 and 5.2. It looks a bit odd to have such an introductory paragraph in a paper. The first sentence is very similar to the first sentence of 5.1 and could be merged with that. The second sentence can be removed. The third sentence can easily go as first sentence in 5.1.**

*Authors:* We agree with this comment. We moved this introductory paragraph to the beginning of section 5.1. The second sentence has been removed.

Comment 20:

**Referee #2: P8L6-7 – Clarification needed:** *"The appearance and structure of buried massive ice are similar to those of englacial ice typically observed at the margin of glaciers, ice caps or ice sheet."*

*Authors:* We modified for *"The appearance and structure of the buried massive ice body are similar to those of englacial ice typically observed at the margin of glaciers, ice caps or ice sheet."*

Comment 21:

**Referee #2: P9L25 – Suggested revision:** *"large stagnant ice blocks could have been covered […]".*

*Authors:* Suggestion accepted.

Comment 22:

**Referee #2: P9L25 – Suggested revision:** *"[…] by glacigenic sediments  at or near the ice margins […]".*

*Authors:* Suggestion accepted.

Comment 23:

**Referee #2: P9L25-29 – What about burial by supraglacial meltout till? I think this is the most common process of stagnant ice burial. Ice bodies just drown in their own sediment load during melt-out. At some point the sediment thickness exceeds the active layer depth and further melting stops.**

*Authors:* We agree that melt-out of supraglacial till from the ice surface is a common process allowing burial and preservation of the ice. However, production supraglacial melt-out till requires in situ melting of a debris-rich glacier ice, typically basal ice, that will become buried by a thick, stable, insulating cover of sediment. The buried glacier described in this paper has a very low sediment content. In addition, in situ melting of basal ice usually produces a laminated to bedded diamicton consisting mainly of silt. This is not the case here. The sediment covering the ice consist of coarse sand and gravel that we interpreted as glaciofluvial ice-contact sediment. We suggest that the sediment characteristics rather indicates the burial of the ice occurred as meltwater streams deposited sediments in direct contact with glacier ice.

Comment 24:

**Referee #2: P10L21 – Suggested revision:** *"Evidence that support the englacial origin of the massive ice are: […]".*

*Authors:* Suggestion accepted.

Comment 25:

**Referee #2: P10L22 – Clarification needed:** *"Bubble-rich ice, with small gas inclusions […].*

*Authors:* The meaning of "small" refers to the size of the gas inclusions, rather than the abundance.

Comment 26:

**Referee #2: P10L27 – Suggested revision:** *"[…] the buried englacial ice  is  strongly depleted in heavy isotopes […]."*

*Authors:* Suggestion accepted.

Comment 27:

**Referee #2: P10L29 – Suggested revision:** *"Although stable isotopes cannot yield information […]".*

*Authors:* Suggestion accepted.

[revised manuscript text omitted]

---

## Author Response (AR3)

Comments by the editor were very enlightening and their suggestions useful; we are grateful for her input. His/her careful reading of the manuscript and his/her good knowledge of the subject-matter allowed providing relevant suggestions and additions to the manuscript. We treat each point raised in detail and with great interest.

Note that the line numbers given in this response refer to the revised version of the manuscript in track changes mode.

**Editor**

Technical corrections (23 Nov 2018)

**Editor: Please check and correct one minor technical detail – please check the numbering of figures, tables thoroughly again. The equations in your appendix need to be labeled with capital letters, for example A1, B1, etc. Appendix C starts with equation (2) instead of C1.**
Authors: We have checked the numbering of all figures and we have corrected the labelling of the equations provided in Appendix B and C.

General comments

Comment 1:
**Editor: Please clarify and correct your results of the water samples, specifically referring to S+ as "cation". S does not form a cation in aqueous solutions, but anions ($SO_2$, $SO_3$, $SO_4$).**
Authors: We agree and removed it from the results and figure 7 since it does not enhance the discussion.

Comment 2:
**Editor: The discussion of the occurrence, formation and processes of buried ice needs to be extended and sharpened to extend beyond the local/regional Bylot Island perspective. Please discuss and distill, especially with available literature available from Russia, Alaska (similarities of occurrence, processes).**
Authors: We added this information (*section 5. Discussion*) to further discuss the occurrence, formation and process of buried ice beyond the regional Bylot Island perspective.

- "*Our interpretation is similar to other studies that reported the burial of glacier ice by glaciofluvial sedimentation (Dallimore and Wolfe 1988; Ingólfsson and Lokrantz 2003; Kaplanskaya and Tarnogradskiy 1986, Belova et al., 2008).*"
- "*The formation of theses ice-cored landforms follows a typical sequence of events where the ice is first buried under a sufficient amount of sediments exceeding the active layer thickness (Lukas et al. 2005, Benn et Evans, 2010). Beyond the active margins of many glaciers and ice sheets, there are broad areas of glacial deposits, till and outwash underlain by glacier ice. Gravitational and glaciofluvial processes are often identified as the most important processes of sediment deposition and reworking that lead to the burial of englacial ice (Benn and Evans, 2010). In basal zones of glaciers and ice-sheets,*"

*in situ melting of stagnant debris-rich basal ice produces a supraglacial sediment cover (i.e. melt-out till) that accumulates on the ice, which inhibits its ablation. Both stagnant englacial and basal ice becomes progressively isolated from the upper active flowing ice. These ice-cored landforms adjust to non-glacial conditions and their evolution is strongly linked with climate-driven processes (e.g. active layer deepening due to warmer atmospheric temperature or active layer detachment slides following heavy rains)."*

- *"Such ice-cored landforms inherited from the partial melting of buried glacier ice are common in formerly glaciated permafrost regions across Northern Canada and Siberia but their spatial distribution remains poorly known (Belova, 2008; Ingolfsson et Lokrantz, 2003; Kokelj et al., 2017; Dyke and Savelle, 2000; Lakeman and England, 2012)."*

Comment 3 (Format issue):
**Editor: Please check and improve font (size and format) of letters in your figures. For examples, "Mittimatalik" is not readable with 100% view. Also, please include sublabels of figures (a, b, c) and refer to these in your figures captions and in your text. Please make sure that the font size of legends and axis labels are large enough (should be readable with 100% view).**
Authors: Modification made.

Comment 4:
**Editor: Please check and improve the language thoroughly. I have made some few suggestions in the text.**
Authors: Suggestions accepted.

Comment 5:
**Editor: Why did you choose a supplement instead of appendix for additional method description? Appendices are part of the manuscript whereas supplements are published along with the manuscript. Since you are referring to a method, I suggest to use the appendix format, since it makes it easier to read. Equations, figures and tables in supplements should be numbered as (S1), Fig. S5 or Table S6. Sections are numbered as S3, S3.1, and S3.1.1.**
Authors: Modification made.

Specific comments

Comment 1:
**Editor: P3L11 - Please add references to sub figures (a,b,c)**
Authors: Modification made.

Comment 2:
**Editor: P4L6 - Add sublabel of figure.**
Authors: Modification made.

Comment 3:

**Editor: P4L8-9 – Sentence unclear, please rephrase**

Authors: We modified for *"The lower and lateral contacts between the ice body and the surrounding sediments could not be delineated"*.

Comment 4:

**Editor: P4L10 – Small letters after enumerations**

Authors: Modification made.

Comment 5:

**Editor: P4L13 + P5L11 – Which years was the field sampling done and when was the laboratory analysis done?**

Authors: We added:

- *"Between 2011 and 2013, ice cores were extracted every 10 cm […]"*.
- *"In 2013, all samples (n=80) were melted […]"*.

Comment 6:

**Editor: P4L29 – I suggest to move it to appendix, not supplementary.**

Authors: We agree and modified for "*[…] is presented in Appendix A*".

Comment 7:

**Editor P4L13 + P6L23 - This is not a cation; do you mean SO2, SO3, SO4? Same with P. Please clarify.**

Authors: See comment #1 (General comments).

Comment 8:

**Editor: P5L13 - Were the samples acidified in the field? Were they kept cold? When was the analysis done? Please add this information in the method section.**

Authors: We agree and we added this information in the method section:

- *"All samples were kept frozen, shipped and stored in our laboratory for further analysis."*
- As for the acidification, all samples were kept frozen prior to shipment to our laboratories. Then, the ice samples were thawed in the laboratory and acidified before processing. We added some details to clarify this part: *"In 2013, all samples (n=80) were melted, filtered (0.45 μm diameter filter) and acidified in the laboratory in sealed polyethylene bottles prior to analyses."*

Comment 9:

**Editor: P5L29 – Classification: Why only pore ice? Could it not be other types of ice? Above you refer to massive ice.**

Authors: In this section, we refer to the sediments (units B and C) overlying the massive ice body. These sediments consist mostly of coarse sands and gravels with a structureless cryostructure made of pore ice. Since this information only given in section 4.2 of the "Results", we modified for *"Additionally, the ground ice in the sediments was analyzed […]"* to clarify this part.

Comment 10:
**Editor: P6L4 - Correct your figure labeling (a or A) throughout text and figures, please.**
Authors: Modification made.

Comment 11:
**Editor: P6L30 - Please clarify: massive, pore, intrasedimental?**
Authors: We modified for "*The D-excess of snow, ice wedge, glacier C-93 and ground ice (unit B) samples averages [...]*". The type of ground ice (pore ice) is defined later in the results (Section 4.2 - Cryostratigraphy and properties of the overlying sediments).
- P7L8: "*[...] has an ice-poor sediment cryofacies with a structureless cryostructure, essentially made of pore ice.*"

Comment 12:
**Editor: P8L5-6 - The table does not include the locations. "Described" in literature should be detailed. Please also add the Russian Literature/locations to give a complete circum-Arctic perspective of occurrence and formation of this ice.**
Authors: We agree and added the locations (location, country) in the table.

Comment 13:
**Editor: P9L8 - Please extend the discussion to a larger circum-arctic perspective by including other regions, especially the Russian/Siberian locations and potentially also burial/preservation processes.**
Authors: See response to comment #2 (General comments)

Comment 14:
**Editor: P9L32 - Be consistent, please use the correct and same symbols throughout the paper (Ca2+,..)**
Authors: Modification made. "*The average concentrations of $Ca^{2+}$, $Na^+$ $Mg^{2+}$ and $K^+$ in units B and C are [...]*"

Comment 15:
**Editor: P12L4-7 - Small letters after enumerations**
Authors: Modifications made.

Comment 16:
**Editor: P10L21-22 - Please add the citation (with doi) of the data in your reference list as well.**

[revised manuscript text omitted]